# Photochemically Induced Changes of Dissolved Organic Matter in a Humic-Rich and Forested Stream

**Christin Wilske [1,*], Peter Herzsprung [2], Oliver J. Lechtenfeld [3], Norbert Kamjunke [1] and Wolf von Tümpling [1]**

[1] Department of River Ecology, Helmholtz Centre for Environmental Research—UFZ, Brückstraße 3a, 39114 Magdeburg, Germany; norbert.kamjunke@ufz.de (N.K.); wolf.vontuempling@ufz.de (W.v.T.)

[2] Department of Lake Research, Helmholtz Centre for Environmental Research—UFZ, Brückstraße 3a, 39114 Magdeburg, Germany; peter.herzsprung@ufz.de

[3] Department of Analytical Chemistry, Helmholtz Centre for Environmental Research—UFZ, Research Group BioGeoOmics, Permoserstr. 15, 04318 Leipzig, Germany; oliver.lechtenfeld@ufz.de

[*] Correspondence: christin.wilske@gmx.de

**Abstract:** Photochemical processing is an important way to transform terrestrial dissolved organic matter (DOM) but was rarely investigated by ultra-high resolution mass spectrometry. We performed an irradiation experiment with water from a shaded forest stream flowing into a lit reservoir. Bacterial activity explained only 1% of dissolved organic carbon (DOC) decline in a combined bacterial and photodegradation approach. Photodegradation decreased the DOC concentration by 30%, the specific ultraviolet (UV) absorption by 40%–50%, and fluorescence intensity by 80% during six days. The humification index (HIX) decreased whereas the fluorescence index (FI) did not change. Two humic-like components identified by parallel factor analysis (PARAFAC) of excitation–emission matrices followed the decrease of fluorescent DOM. Changes of relative peak intensities of Fourier transform-ion cyclotron resonance mass spectroscopy (FT-ICR MS) elemental formula components as a function of cumulated radiation were evaluated both by Spearman's rank correlation and linear regression. The FT-ICR MS intensity changes indicate that high aromatic material was photochemically converted into smaller non-fluorescent molecules or degraded by the release of $CO_2$. This study shows the molecular change of terrestrial DOM before the preparation of drinking water from reservoirs.

**Keywords:** photodegradation; DOC; $SUVA_{254}$; 2D fluorescence; HIX; FT-ICR MS; PARAFAC; light-dark

## 1. Introduction

The importance of photodegradation of dissolved organic matter (DOM) and of absorbance and fluorescence efficiency has been documented since 1981 [1–4]. Since 1995 different water bodies were investigated because of the important effect of the DOM photochemistry in aquatic environments. Recent studies have shown that land uses affect the amount, quality and sources of DOM in streams and rivers e.g., [5,6]. Other studies investigated the influence of different forest types on the DOM characteristics [7–11]. In several lakes and streams world-wide an increase of dissolved organic carbon (DOC) was observed [12,13]. Higher DOC concentrations and more humic substances in the water bodies resulted in higher photooxidation rates [3,14]. Furthermore, lake DOC is less photoactive than stream DOC [15].

Production, discharge, decomposition and quality changes of dissolved organic carbon in pre-dams of reservoirs was described by [16] with the focus on biological activities. Such pre-dams are often shallow and flooded with light. Hence, beyond long term (slowly) biological activities short term photochemical processes might have an influence on the DOM quality change. Ultraviolet (UV)

radiation can have several effects, it bleaches colored water (decrease in the absorptivity of the DOM in the UV and visible spectral regions) and high molecular material is decomposed into low molecular weight organic compounds, carbon gases, unidentified bleached organic matter and nitrogen- and phosphorus-rich compounds [17–20]. The behavior by UV irradiation of humic-rich water bodies and also of Sigma-Aldrich standards have been studied since 1994 [3]. They also investigated that by irradiation at 254 nm for 12 h, 95% of fulvic-like, and for 58 h, 95% for humic-like material, decreased. The average molecular weight was determined by capillary electrophoresis and decreased from 1800 to 300 g/mol [3]. Degradation of larger and photoreactive molecules into smaller and more UV-transparent compounds determined by gel filtration was also studied by [2].

DOC consists of a large variety of different organic components, depending on the sources. It can be built by autochthonous, in-lake processes or enter from allochthonous water and soil sources from catchment areas [21]. Allochthonous DOC contains often highly colored humic and fulvic acid material, while autochthonous DOC which was generated by phytoplankton or macrophytes is less colored [22]. Often DOC is influenced by microbial mineralization. A substantial fraction (over 50%) of the terrestrial derived DOC was mineralized [23,24] and (Caraco in [25]). The specific UV absorption (SUVA) as a bulk optical parameter was used for the characterization of DOM quality in soil water [26] and stream water [27]. The $SUVA_{254}$ is a good indicator of carbon aromaticity and provides an estimate of the average aromatic content for all DOM compounds present in solution [28–30]. Additionally it is useful to assess the removal efficiency of DOM with coagulation [31] and it was described to be a useful proxy for molecular weight [32].

The determination of fluorescent matter (fDOM) using the 2D fluorescence spectroscopy represents one possibility to identify main characteristics of the DOM with "low-cost" and simple analytical instrumentation and was used for more than a decade to characterize organic material [33–36]. Because of the simple and low cost method, it is increasingly applied to assess DOM quality (search function of ScienceDirect queried on 12.12.2019 gave 671 research articles with PARAFAC (parallel factor analysis) and DOM as keywords since 1996). The excitation–emission matrices (EEMs) have been used for the calculation of the fluorescence indices and PARAFAC components. EEMs provide information about DOM sources, reactivity ecological function or chemistry [37–41]. [37,38] used peak picking techniques but in current researches PARAFAC is used to define fluorescent components [42] and are commonly attributed to properties of DOM like e.g., humic-like, fulvic-like or terrestrial [35,43–45].

DOM probably consists of thousands or even millions of different molecules and the identification of the isomeric structure of each molecule is still far from any instrumental analytical realization. The highest analytical resolution of DOM can be achieved by Fourier transform-ion cyclotron resonance mass spectroscopy (FT-ICR MS) [46–50]. However, the analysis of components using such sophisticated analytical methods is an expensive task. Few studies combined FT-ICR MS with bulk parameters. Reference [51] used Spearman's rank correlation coefficients (normal distribution is than not a boundary condition) to combine FT-ICR MS-derived molecular formula intensities with humic-like fluorescence intensities. Also reference [8] used the rank correlation coefficients to determine the molecular formulas associated with fDOM. Reference [6] used the Pearson correlation to relate bulk properties and bio- and photo-reactivity to the relative abundance of molecular formulas. Other studies combined these formula intensities with the fluorescence indices [52,53]. A recent study [54] related the relative peak intensities from FT-ICR MS with reactive species concentrations like hydroxyl radicals and singlet oxygen using the Spearman's rank correlation. In our study we present a new approach, the relation of solar radiation to the FT-ICR MS derived molecular formula intensities (and other bulk parameters like DOC, SUVA, fluorescence indices). As yet, global radiation has been combined and presented only with parameters different from DOM quality. Published in 2015, differences between summer 2011 and 2012 and between streams or ponds (light-limited/flow conditions) were related to global radiation using meteorological data [55]. Here, we correlated DOM quality parameters to cumulated solar radiation (see methods Section 2.8.4).

We tested the hypothesis (4.1.) that the composition of DOM after irradiation with natural sunlight is different in the 0.2 µm filtered water (no abundance of bacteria assumed) and in the 2 µm filtered water (presence of bacteria suggested). We hypothesize that bacteria will further degrade a part of arising photo products.

Because of the relative high resolution of DOM variations as function of cumulated global radiation we hypothesize (4.2.) that different photochemical reaction behavior will be detectable, photo products, photo degraded components, and components with variable abundance changes.

As prediction for the photochemical DOM change we hypothesize (4.3.) that photoproducts will be more aliphatic (loss of DBE), more oxygen-rich (oxidation of DBE) and less molecular weight (braking bonds, loss of $CO_2$ by decarboxylation) compared to photo degraded components.

We suggest (hypothesis 4.4.) that DOM quality changes will be visible both by characterization of bulk parameters (DOC, SUVA, EEMs, fluorescence indices, PARAFAC components) and of solid-phase extraction of dissolved organic matter (SPE-DOM) components as analyzed by FT-ICR MS.

## 2. Materials and Methods

### 2.1. Study Area and Sampling

Water was collected in July 2018 from the inlet of Rote Mulde into the pre-dam of a drinking water reservoir located in Saxony East Germany (Lat: 50.401847/Lon: 12.380528) at an altitude of 770 m above sea level. The Rote Mulde stream is 98% forested and has a humic-rich character. The inlet of the pre dam was sampled because of the forested landscape and the transition from shade to light, whereby only little photochemical activity was assumed to be present until the sampling.

The surface water sample (10 L) was filtered through membrane filters with a pore size of 2 µm and stored at 4 °C. The first filter step was assumed to remove particulate organic matter. An aliquot of the first filtrate was subsequently filtered through 0.2 µm membrane filter and stored in the refrigerator. In the second filtering step, the bacteria were removed.

### 2.2. Experimental Design

Round-bottom quartz flasks have been used, so that the largest possible surface area of the contained water is irradiated with sunlight, to exclude shadows and reflections, according to [56]. The round-bottom flasks were placed (both groups) with about 1.5 m space in between on the flat house roof to exclude shadows of one to other placed flasks. The space between the different triplicates was about 30 cm. Air exchange in the flasks was ensured and the global radiation was logged. On 31 July 2018 the experiment began with the sunrise at 5:30 am. At the first two days, water samples alternating from the replicates were taken after two, four, eight and sixteen hours (20 mL for DOC, 2D fluorescence and bacterial production; 50 mL for the FT-ICR MS measurements). The total experiment runs for 6 d (for details see Table S1).

The global radiation was measured with a LI-1400 dataLogger of LI-COR (LI-COR, Lincoln, NE, USA). Every 5 min the average was logged. The single values ($W/m^2$) were summed over the entire time period.

### 2.3. Bacterial Productivity

The samples of the experiment with 2 µm filtered water samples were collected for measuring the bacterial production at 9 different sample points (0 $kW/m^2$, 10.0 $kW/m^2$, 43.3 $kW/m^2$, 83.3 $kW/m^2$, 119.8 $kW/m^2$, 156.3 $kW/m^2$, 168.7 $kW/m^2$, 282.6 $kW/m^2$ and 436.2 $kW/m^2$ cumulated radiation (cumRAD)). Additionally, two samples of the 0.2 µm filtered water were collected at the beginning and at the end of the experiment as a control.

Production of suspended bacteria was measured using the leucine technique [57] as described by [58]. Triplicate 5 mL aliquots and one formalin-treated control (3.7%, final concentration) were spiked with 14C leucine (12.2 MBq µmol$^{-1}$, Sigma, 50 nM final concentration). Samples were incubated

in the laboratory at in situ temperature for 1 h in the dark. Proteins were extracted in trichloracetic acid (5%, final concentration) and filtered onto 0.2 μm nuclepore membranes. Radioactivity was measured using a liquid scintillation analyzer (MicroBeta Trilux 1450 LSC, WALLAC, Turku, Finland). Carbon production was calculated using the equations of Simon and Azam [57].

*2.4. Dissolved Organic Carbon and Specific Ultraviolet (UV) Absorbance at 254 nm (SUVA$_{254}$)*

The DOC concentration was measured with a DIMATOC 2000 (Dimatec Analysentechnik GmbH, Essen, North Rhine Westphalia, Germany) according to DIN EN 1484:1997-08 [59].

For the description of the absorbance of ultraviolet light in a water sample at a specified wavelength (254 nm) and normalized for DOC concentration, we calculated the SUVA. It was calculated using the decadic light absorption coefficient at 254 nm (m$^{-1}$) (measured with the Aqualog® (HORIBA Jobin Yvon GmbH, Bensheim, Germany)) normalized to DOC (mg C L$^{-1}$). The SUVA is highly correlated to hydrophobic organic fraction of DOM [60]. Results with same SUVA may differ in their distribution around the average value [61] and consequently in their reactivity [52]. Typically a higher SUVA is associated with greater aromatic content [29]. SUVA > 4 indicates mainly hydrophobic and especially aromatic material, values < 3 illustrates mainly hydrophilic material [62].

*2.5. Two-Dimensional (2D) Fluorescence Spectroscopy*

The excitation-emission-matrices (EEMs) were acquired on all filtered water samples in a 1 cm quartz cuvette using the Aqualog® at excitation range 240 to 600 nm with 1 nm increments and emission range 212 to 621 nm with 0.42 nm (1 pixel) increments. The integration time was 1 s. All measured EEMs were blank-subtracted (Milli-Q Water), corrected for instrument-specific biases and inner filter effects for each measurement (time point). Additionally, the EEMs were normalized to the Raman and Rayleigh (first and second radiation) area (10 nm) of pure water.

*2.6. Parallel Factor Analysis (PARAFAC)*

The classical factor analysis was developed at the beginning of the 20th century and is essential to gain information from two-dimensional data matrices. Since 1970, three-dimensional data structures can be decomposed e.g., by PARAFAC developed by [63].

The PARAFAC decomposition has several modeling steps. At first the outliers are determined in the data set. Outlier should be removed from the data. Following, the statistical tests have to be repeated. During the outlier test models are created where a number of components can be chosen. The components were calculated iterative. After selecting the number of components $q$, the elements $a_{(i \times f)}$, $b_{(j \times f)}$, $c_{(k \times f)}$ of A,B and C matrices are calculated by the alternating least squares algorithm (ALS algorithm). Each element of the PARAFAC model (decomposes the $X_{ijk} = \{x_{ijk}\}$ data matrix) can be described by the following formula [64]:

$$X_{ijk} = \sum_{f=1}^{q} a_{if} b_{jf} c_{kf} + e_{ijk}, \; where \; (i = 1, \ldots, n; j = 1, \ldots, m; k = 1, \ldots, p) \tag{1}$$

where $f$ is the fluorescence intensity, $i$ the number of samples, $j$ the emission wavelength and $k$ the excitation wavelength. The next step in the PARAFAC decomposition is the explorative data analysis, where the core consistency, the explained variance and the sum of residuals were checked. If the explained variance is 100%, then the model is described solely by the main components. If it is lower than 100%, then the model has a residual matrix with the elements $e_{ijk}$.

After exploratory data analysis, a new PARAFAC model is formed, which can be validated by the split half validation of Stedmon [65]. It can be checked whether the model is valid, with the selected number of components [36]. A more detailed description of the multi-way decomposition method for PARAFAC is given by [66,67].

The fluorescent DOM (fDOM) components were identified by PARAFAC using the software SOLO (Eigenvector Research, Inc., Manson, WA, USA). A PARAFAC model was developed using 77 emission–excitation-matrices (EEMs). The PARAFAC model was validated using split half validation, core consistency diagnostic and visual inspection of the residuals.

*2.7. Solid Phase Extraction (SPE) and Fourier Transform-Ion Cyclotron Resonance Mass Spectroscopy (FT-ICR MS) Measurements*

For analysis of DOM composition via ultra-high resolution mass spectrometry, samples of each time point and treatment were filtered through glass fiber filters (Whatman GF/F) and acidified to pH 2 with HCl. Solid-phase extraction was performed on acidified subsamples of $(0.2 \pm 0.01)$ mg C $((28 \pm 3)$ mL, $n = 77)$ with 50 mg styrene-divinyl-polymer type (PPL) cartridges (Agilent, Waldbronn, Germany) using an automatic sample preparation system (LCTech Freestyle, Obertaufkirchen, Germany) to desalt the sample for subsequent electrospray ionization (ESI)-MS. The SPE-DOM was eluted with 1 mL methanol (Biosolve, Valkenswaard, Netherlands) and was kept frozen until analysis (for further details see [49]). The average carbon-based recovery of DOM was $(51\% \pm 11\%)$. SPE-DOM samples were diluted to 20 ppm and mixed 1:1 (*v/v*) with ultrapure water immediately prior FT-ICR MS analysis.

An FT-ICR mass spectrometer equipped with a dynamically harmonized analyzer cell (solariX XR, Bruker Daltonics Inc., Billerica, MA, USA) and a 12 T refrigerated actively shielded superconducting magnet (Bruker Biospin, Wissembourg, France) instrument located at the ProVIS Centre for Chemical Microscopy within the Helmholtz Centre for Environmental Research was used in ESI negative ionization mode (capillary voltage: 4.5 kV). Extracts were analyzed in random order with an autosampler (infusion rate: 10 $\mu$L min$^{-1}$). For each spectrum, 256 scans were co-added in the mass range 150–3000 *m/z* with a 4 MWord time domain. Mass spectra were internally calibrated with a list of peaks (250–640 *m/z*, $n > 129$) commonly present in terrestrial DOM and the mass accuracy after internal calibration was better than 0.2 ppm. Peaks were considered if the signal/noise (S/N) ratio was greater than four.

*2.8. Calculations*

2.8.1. Indices Humification Index (HIX), Biology or Freshness Index (BIX) and Fluorescence Index (FI)

We calculated the humification index (HIX = ratio of areas under the emission curve at 435–480 nm and 300–345 nm plus 435–480 nm at an excitation wavelength 245 nm [68]), the fluorescence index (FI = emission intensity at 470 nm divided with that of 520 nm at 370 nm excitation [39]) and the biology or freshness index (BIX or $\beta/\alpha$ = ration of emission intensity at 380 nm and maximum intensity between 420 and 435 nm at an excitation wavelength of 310 nm [69]) using Python 3.7.1 (Python Software Foundation, Beaverton, OR, USA) (Table S2). The HIX describes the degree of humification (high HIX = more humifical material), FI the kind of source (microbial FI $\approx$ 1.8 or terrestrial FI $\approx$ 1.3) and BIX the age (high BIX = larger contribution of more freshly produced DOM) [52].

2.8.2. Statistical Analysis of Bulk Parameters

For the comparison of the two different filter levels the Spearman's rank correlation was calculated. The ranks for DOC, SUVA, fluorescence indices (FI, HIX, BIX) and the two PARAFAC components were calculated and are listed in Table S3.

### 2.8.3. Identification of Photochemical Reaction Behavior of Molecular Formula Components

As shown in Table S2, 77 single FT-ICR mass spectra had to be recorded. 13 different time points multiplied by each three replicates and two different filters and one replicate was lost by one bottle fall over ($[13 \times 3 \times 2] - 1 = 77$). For each of the 77 samples (representing both 0.2 and 2 μm filtration) the relative mass peak intensity was calculated (intensity of component divided by the sum of all components as described by [70] or [71]. The presence count was calculated (component present in 1–77 samples). For statistical calculations only components present in all 77 samples were used (total common presence [51].

For each reaction time point (with assigned cumulated global radiation) the median relative intensity was calculated from each three replicates (for sample t = 11 (0.2 μm) from duplicate). All further calculations originated from these 13 (median samples, representing either the 0.2 μm or the 2 μm filtration). All molecular formula data (median relative intensities) can be found in the supporting information (SI_data_base.xlsx).

### 2.8.4. Differentiation between Photo Products and Photo Degraded Components

Components with increasing relative intensity as function of time (and cumulated global radiation) are photo products, components with decreasing intensity are photo-degraded components. The significance of intensity change with time was first calculated in the most possible robust way by Spearman's rank correlation (as introduced by [51] for molecular formula components) for relation of DOM quality with cumulated solar radiation in order to search for three different types of reactivity, photo products (positive correlation), degraded components (negative correlation), non-reactive components (no correlation). As pre-arrangement for rank correlation the intensity ranks for all 25,987 ($1999 \times 13$) components (present in all 77 triplicate samples) in all 13 (median intensity) samples were calculated. The intensity ranks were used for calculation of the inter sample ranks (ranking of each component in 13 samples) as described by [51,72] and by [73]. The inter sample ranks can be used for calculation of rank correlation component wise with any external parameter. Here we correlated each component median intensity inter sample rank with the ranking of cumulated global radiation. We suggested and confirmed by calculation that cumulated global radiation increased with increasing reaction time. So the last time point received simply the first rank (highest cumulated global radiation) and the first time point the 13th rank (no global radiation; Table S4). A positive sign of the rank correlation coefficient $r_s$ means photo products, a negative sign photo degraded components. The level of significance as calculated from the absolute value of $r_s$ and the number of samples (Table S8, [74]) serves as measure of the probability of reaction behavior. High levels of significance mean high probability for components to be reactive, lower levels mean less probability for reactivity and no significance means that a component is probably not reactive.

A second possibility for quantification of the reactivity is the calculation of a linear regression (median relative intensities of a component versus the cumulated global radiation at the 13 time points (Figure 1)). The slope and the coefficient of determination are both received from this calculation as a measure for the reactivity of components. The calculation procedure is described by [50]. We did not use the changes in relative intensities to calculate mass turnover (i.e., compound wise concentration changes) in the samples, as response factors for the compounds are unknown and may be affected by changes in overall sample composition. The calculated slopes were used as a semi-quantitative value, to be able to evaluate the steepness of the slopes and the largest shift in intensity. Positive slopes reflect products (total or in part) and negative slopes reflect totally or partly degraded DOM components.

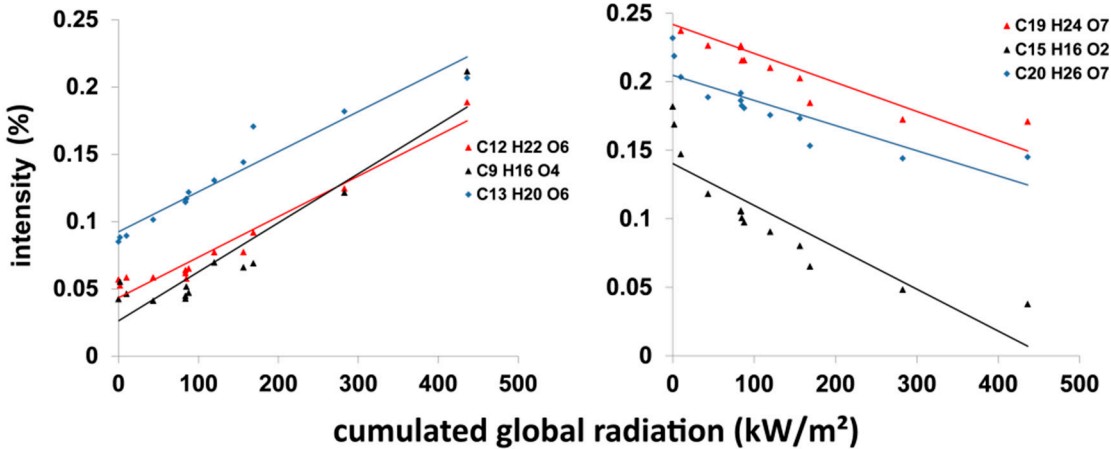

**Figure 1.** Linear regression of normalized intensity (%) of specific molecular formulas over the cumulated global radiation (kW/m²) as an example.

## 3. Results

### 3.1. Solar Radiation

The monthly upscaled measured value of the cumulated global radiation (cumRAD) was about 163 kWh/m². The calculated monthly sum for August 2018 of the Deutsche Wetterdienst (DWD) was between 156 and 160 kWh/m² [1].

### 3.2. Bacterial Production

As expected, no planktonic bacterial production has been observed during the experiment with the 0.2 μm filtered water.

In the 2 μm experiment, the planktonic bacterial production increased to about 6 μg C L⁻¹ d⁻¹ after 4 h but decreased again after another 4 h and stayed at a low level for the rest of the experiment (Figure 2e). The short-term peak resulted in total production of 1 μg C L⁻¹, representing a bacterial organic carbon demand of 10 μg C L⁻¹ by assuming 10% bacterial growth efficiency. This carbon demand may explain only 1% of the DOC decrease in the experiment (>1 mg C L⁻¹).

To prove that the general behaviour of both series without and with bacteria is comparable the Spearman's rank correlation coefficients with the inter sample ranks of the median of the measured parameter have been calculated (Table 1).

**Table 1.** Spearman's rank correlation coefficients of the calculated ranks of the median of the observed parameter (humification index (HIX), dissolved organic carbon (DOC), specific UV absorption (SUVA) and biology index (BIX)) between 2 μm and 0.2 μm filtered water samples. It is sorted by good match (high $r_s$).

| Parameter | HIX | Scores Comp 2 | DOC | SUVA$_{254}$ | BIX | Scores Comp 1 |
|-----------|-----|---------------|-----|--------------|-----|---------------|
| $r_s$ | 0.98 | 0.93 | 0.88 | 0.86 | 0.83 | 0.60 |

Very high coefficients describe the very similar course of both series excluding the scores of the first PARAFAC component. Therefore only the results of the pure photochemical experiment (filtration < 0.2 μm) and the difference in Score Comp 1 are described in detail and all results and figures of the experiment with a filter level of 2 μm are shown in the supporting information.

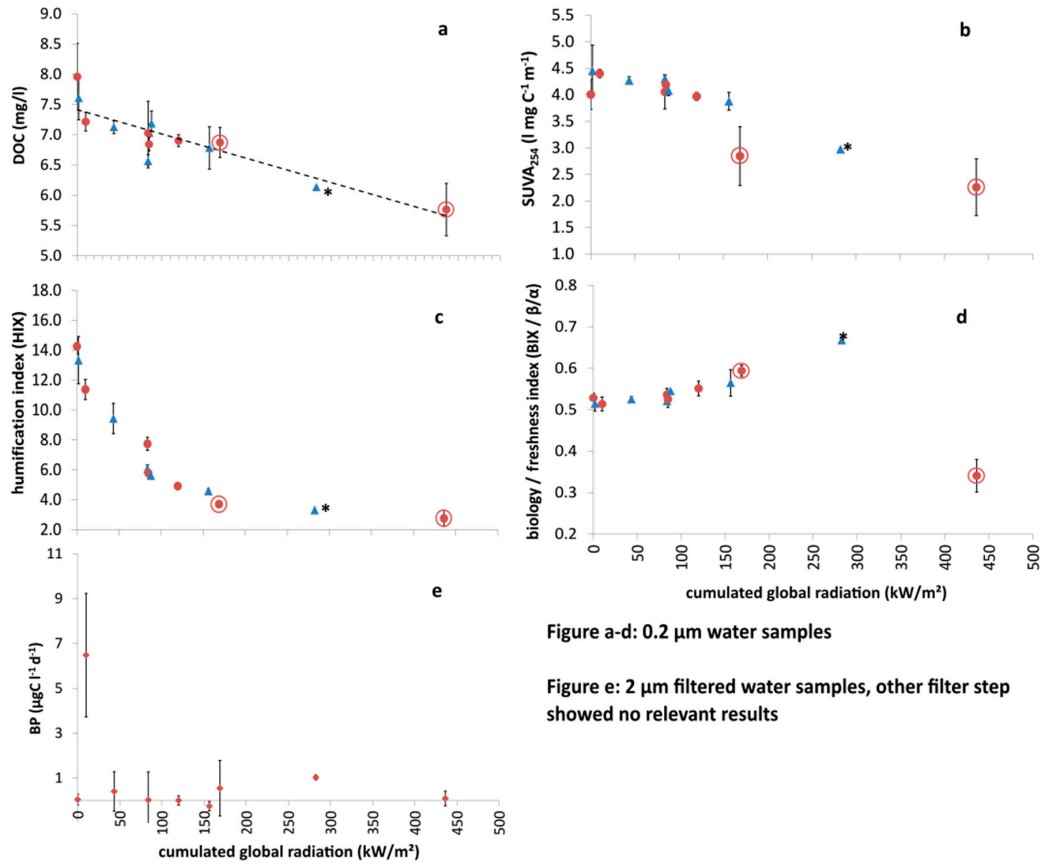

**Figure 2.** Mean values and the standard deviations of DOC (**a**), the specific UV absorbance at 254 nm (SUVA$_{254}$) (**b**), the humification index (**c**), the freshness index (**d**) vs. cumulated global radiation of the 0.2 μm filtered water and (**e**) the dynamic of planktonic bacterial production vs. cumulated global radiation of the 2 μm filtered water. The circular red points represent the averages of the three replicates of Rote Mulde (RMU) 1–3 and the triangular blue ones the replicates of RMU 4–5. The dotted line in (**a**) represents a regression line r$^2$ = 0.81. Dots marked with * are median of the water samples and had no standard deviations, because there were only 2 instead of 3 replicates.

*3.3. Changes in Dissolved Organic Carbon (DOC)*

The irradiation of the water samples resulted in a decrease of DOC concentration during the photochemical process (Figure 2a). 20% DOC was lost after two days (150 kW/m$^2$ cumRAD or 40 h) 30% at the end.

*3.4. Changes in SUVA$_{254}$ (L mg C$^{-1}$ m$^{-1}$)*

The water samples had a high proportion of coloured aromatic compounds, as indicated by SUVA$_{254}$ (Table S2). Like DOC, the specific adsorption at 254 nm (SUVA$_{254}$) decreased during the irradiation. After two days (150 kW/m$^2$ cumRAD or 40 h) only 3% of the value of the SUVA$_{254}$ decreased but at the end the loss was between 40% and 50% (Figure 2b). At 168 kW/m$^2$ cumRAD was a sudden drop.

*3.5. Changes in the Excitation–Emission Matrices (EEMs)*

The irradiation of the water samples resulted in a loss in humic-like fluorescence. The highest decrease was detected during the first day (Figure 3a). On the first day the fluorescence intensity decreased about 40%–50% in a big region of the excitation range from 240 to 280 nm (peak A) and 320 to 400 nm (peak B) and the emission range from 380 to 500 nm. During the first night, the fluorescence intensity increased a little bit (about 5%) in a region of the excitation range from

280–320 nm and the emission range from 380 to 460 nm (peak C). On the second day, this region was reduced again from 9:30 am. Nothing significant happened in the morning from 5:30 am to 9:30 am (Figure S11f,g). During the following days (Figure 3c,d), the fluorescence intensities decreased mainly about 20% in the region of peak A. In Figure 3e the difference between 282 and 436 kW/m$^2$ cumRAD is shown. At this time, the fluorescence intensity decreased mainly (about 25%) in a region of the excitation range of 250 to 320 nm and an emission range of 290 to 380 nm (peak D). This third component could not be calculated validly with the PARAFAC model. Figure 3f describes the total decrease of the fluorescence intensity in the 0.2 µm filtered water sample (about 50% peak B and about 75% peak A).

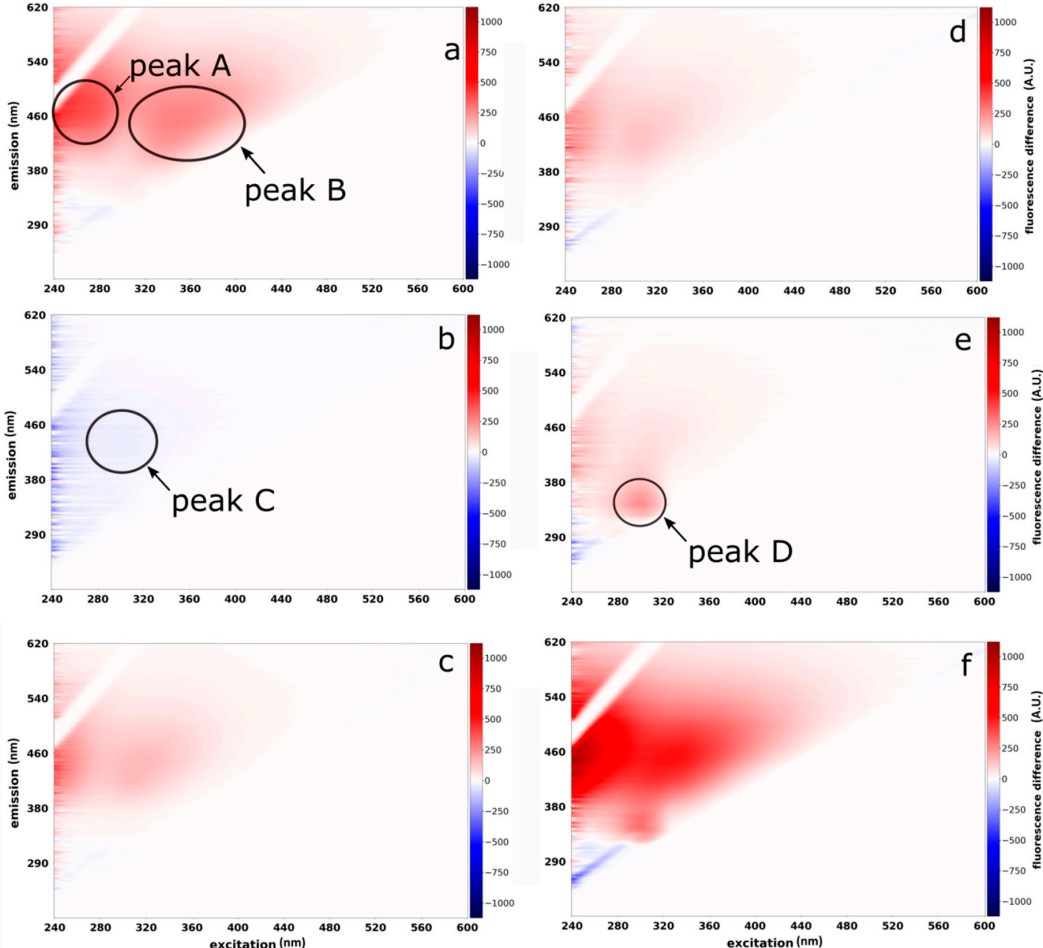

**Figure 3.** Fluorescence difference spectra of the median of the different samples and a filter level of 0.2 µm. In (**a**) the difference of the first day and (**b**) the difference overnight of the first and the second day (no radiation) is shown. The decrease of the fluorescence of the second day is shown in (**c**). In (**d**,**e**) the differences of the third day and between days 4–5 are given. The figure (**f**) shows the total difference of the experiment for the 0.2 µm filtered water sample. The white bar results from the correction of Rayleigh scattering.

### 3.6. Changes in Fluorescence Indices

There was no significant decrease or increase of the fluorescence index (FI) observed with values of about 1. The FI for both experiment series are shown in the supporting information (Figure S8a,b).

There was a decrease of the humification index (HIX) within the time series (Figure 2c). After two days (150 kW/m$^2$ cumRAD or 40 h) between 60% and 70% was removed. Totally the HIX decreased about 80%. Overnight was a sudden drop of the HIX.

The freshness index (BIX or β/α) showed a rising trend up to 282 kW/m² cumRAD. Only the last measured value was below the start value. The total decrease was between 30% and 40% of the starting value (Figure 2d).

### 3.7. PARAFAC Model of the Experiment

The fluorescence indices showed that a small area of the fluorescence DOM (fDOM) has decreased. To study the behavior of the entire fDOM, a PARAFAC analysis was performed. The region of the most reduced fluorescence intensity can be described by two calculated PARAFAC components. The model was valid, which was proofed with a split half validation (Figure S10) and the halves agree to 99.2% as well for the Mode 2 (emission loadings) as for the Mode 3 (excitation loading). The reduced part, described by peak D (Figure 3d), indicated a third PARAFAC component but this could not be calculated as valid. The PARAFAC components describe only 50% of the total model.

Both components decreased until the end of the experiment, which is reflected by the scores (Figure 4). The first component decreased up to 70% and was highly correlated to the SUVA$_{254}$ (Figure S13a; $r_s$ = 0.747). The second component decreased up to 90% and was correlated to the HIX (Figure S13b; $r_s$ = 0.984). In the experiment with a filter level of 0.2 μm, the first component decreased by about 60% in the first two days and totally over 70%. The second component decreased nearly 80% (168 kW/m² cumRAD) and increased again about 15% to 20% until the end.

For the characterization of the PARAFAC components we looked at the emission and excitation loadings of the two components (Figure 5). The blue curve is the first and the green is the second component. The emission maximum of the first component was 420 nm and of the second PARAFAC component about 480 nm (Figure 5a). The excitation maxima were 250 and 310 nm (first component) and 260 and 360 nm (second component: Figure 5b). So, both are humic-like components.

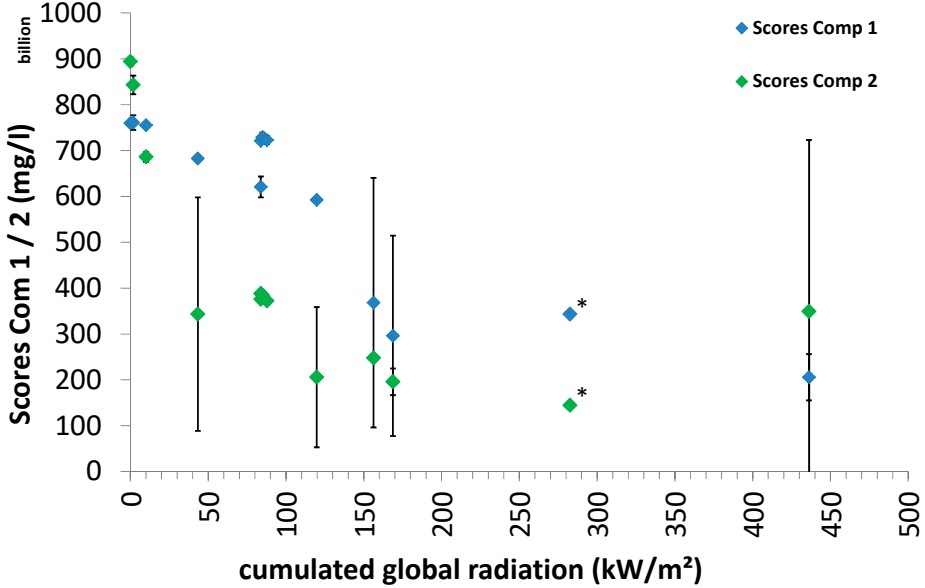

**Figure 4.** Scores of the first and second component (filter level 0.2 μm) of the calculated parallel factor analysis (PARAFAC) model depended on the cumulated global radiation. The blue points are the scores of the first and the green of the second component. Dots marked with * are median of the water samples and had no standard deviations, because there were only 2 instead of 3 replicates.

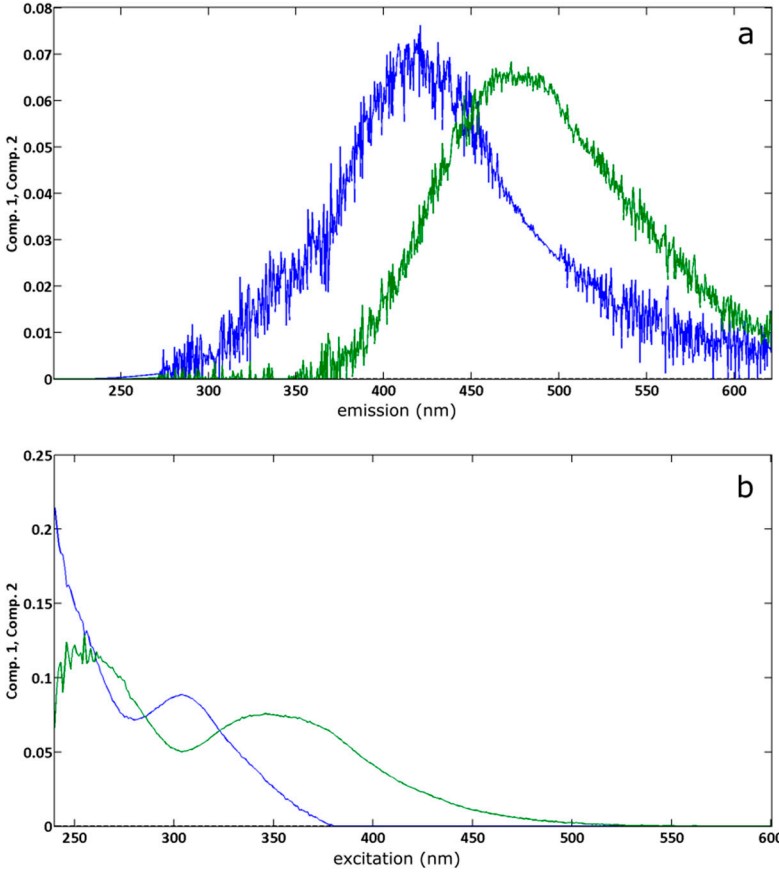

**Figure 5.** Emission (**a**) and excitation loadings (**b**) of the first component (blue) and the second component (green).

*3.8. Photochemical Reaction Behaviour Derived from FT-ICR MS Measurements*

Intensity weighted average parameters are shown in Table S7. The average H/C ratio increased from 1.16 to 1.30, the average O/C ratio was almost constant over the experiment (minimum value: 0.429, maximum value: 0.442), the average mass dropped from 362.1 to 347.6 Da. The nominal oxidation state of the carbon (NOSC) changed from −0.269 to −0.404. The average DBE decreased from 8.56 to 7.16. The average number of carbon (from 18.03 to 17.18) and oxygen atoms per molecule (from 7.6 to 7.21) decreased, while the average number of hydrogen atoms increased (from 21.05 to 22.16). The average number of nitrogen atoms per molecule remained constant while sulphur slightly increased.

The Spearman's rank correlation (for each combination inter sample ranks/cumulated global radiation ranks) is displayed in Figure 6 exclusively for 1540 CHO components. On the first view photo products showed higher H/C values compared to photo degraded components in accordance to the average H/C ratio. The majority of photo products had lower molecular masses.

The van Krevelen and the mass-edited H/C ratios diagrams of the CHO, CHNO and CHOS compounds are shown in more detail in the supporting information. The classification of the Spearman's rank correlation coefficients was undertaken according to the uncertainties ($p < 0.001$; $p < 0.01$ and $p < 0.05$) and is shown in Table S5.

The detailed visualization showed the distribution of DOM molecular formulas (Figures S1 and S2). 867 CHO components with the most significant correlations ($r_s$-values >0.7921 or <−0.7921) were represented by red and black dots, which were calculated with a $p$-value <0.001. 313 CHO components (red dots) had a positive correlation and were found with $1 < H/C < 2$ and $0.2 < O/C < 0.8$ (Figure 6 and Figure S1a). These photo products had a molecular weight between 200 and 500 Da. 554 CHO components (black dots) had a negative correlation and were found with $0.5 < H/C < 1.5$ and an O/C

ratio between 0.2 and 0.8. The molecular mass was between 200 and 600 Da. A distinct region (with H/C approx. 1 and $0.4 < O/C < 0.6$) was related to products with low $r_s$-values between 0.478 and 0.643 (+/−) and the least photo active components (no relevance for DOM-quality change) with $r_s$-values between 0.478 and −0.478 (Figure S1c,d). The highly positively correlated CHO components (red dots) had mainly low molecular weight (only few components with Da > 400). The highly negatively correlated components (black dots) represented to a greater extent high molecular weight components with masses > 400 Da.

A second class had $r_s$-values between 0.7912 and 0.6429 (+/−) and corresponded to an uncertainty of $p < 0.01$. This range contained 170 components. 62 positively correlated CHO (products, orange dots) were found with H/C ratios about 1, with a few components with H/C up to 2 and an O/C ratio between 0.2 and 0.8. These DOM components had a molecular weight between 200 and 650 Da, also the components with a negative correlation. 108 CHO components (blue dots) were found with $0.5 < H/C < 1.5$ and $0.2 < O/C < 0.8$. We observed the same gap in the blue points for the region with H/C about 1 and $0.4 < O/C < 0.6$.

The last separate part contained 125 components and had $r_s$-values between 0.478 and 0.6429 (+/−) and corresponded to $p < 0.05$. The products (yellow dots) represented 46 CHO components with moleculare weights between 200 and 600 Da and with an H/C ratio between 1 and 2. The negative part (green dots; 79 degraded components) were found with $0.5 < H/C < 1.5$ and mostly with $0.2 < O/C < 0.8$.

The range of $r_s$-values from −0.478 to 0.478 was defined as least photoactive (no relevance for DOM-quality change in the investigated environment) and contained 378 CHO components (grey dots). These components had a mass between 200 and 600 Da and were found mainly with van Krevelen coordinates $0.8 < H/C < 1.5$ and $0.3 < O/C < 0.7$. Most of these components were located in the gap of the black dots (Figure S1a, H/C about 1 and $0.4 < O/C < 0.6$) and had a mass of 500 Da.

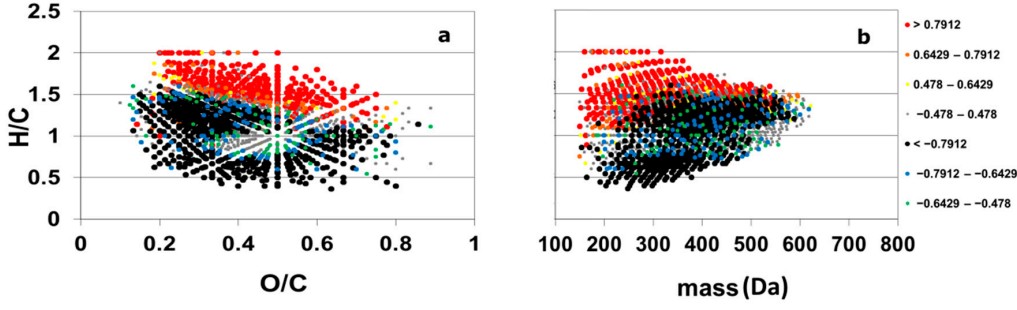

**Figure 6.** Van Krevelen diagrams (**a**) and mass-edited H/C ratios (**b**) of median slope values (triplicates of filter level 0.2 µm) over the cumulated global radiation. Only the CHO-components were selected, which were found in all 77 samples. The color and dots size are related to different ranges of slope values. In the legend, the Spearman's rank correlation coefficient $r_s$ (+/−) is plotted of three uncertainties ($p < 0.001$; $p < 0.01$ and $p < 0.05$).

The rank correlation provided results if a component is a photo product or degraded and if it is significant. The calculation of slopes via linear regression showed in a semi-quantitative manner if a component contributes to overall spectral intensity change. The corresponding coefficient of determination gave information if the intensity change was rather monotonous or not. Finally, more information about the type of photochemical reaction behavior could be received by plotting the relative intensities of a component versus the accumulated global radiation. In the following examples of different types are illustrated: Photo products with more or less monotonous intensity increase (Figure 7a), photo-degraded components with more or less monotonous intensity decrease (Figure 7c and $C_{15}H_{16}O_2$ in Figure 7b), components with both increasing and decreasing intensity within the experimental time window (Figure 7b).

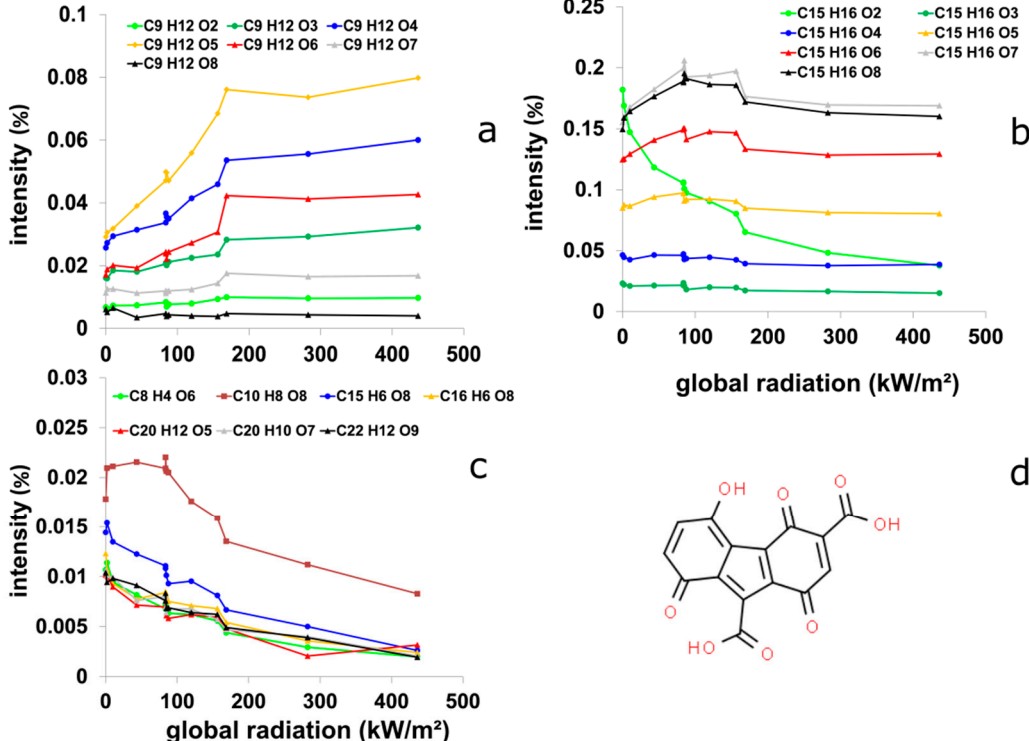

**Figure 7.** Homologous O—series with mainly monotonous increasing intensities versus cumulated radiation (**a**), homologous O—series with different intensity variation (**b**) and homologous O—series with mainly monotonous decreasing intensities (**c**). Possible structure (**d**) for $C_{15}H_6O_8$ present in ChemSpider.

In Figure 7, different DOM components with corresponding slopes are shown. Seven members of a homologous O (number of oxygen atoms) series showed an increasing course over time (Figure 7a). Until the time t = 10 (168 kW/m$^2$ cumRAD), the relative intensity increased, then it remained nearly constant. Of this series, only the molecular formula $C_9H_{12}O_8$ showed no monotonous relative intensity change during the experiment. Another homologous O– series showed different behaviour of the members. In contrast $C_{15}H_{16}O_2$ showed decreasing relative intensity (Figure 7b). The other members showed more or less first increasing and afterwards decreasing intensities. Figure 7c shows components with strong decreasing intensity. For the molecular formula $C_{15}H_6O_8$ isomeric structures were searched using ChemSpider. Only one matching structure (Figure 7d) was found. In Table 2 the calculated slopes of the example components are listed. $C_9H_{12}O_5$ showed the strongest intensity increase and $C_{15}H_{16}O_2$ the strongest decrease.

**Table 2.** Examples of the slope values and the coefficient of determinations of the molecular formulas of Figure 7.

| | a | | | b | | | c | |
|---|---|---|---|---|---|---|---|---|
| Comp. | Slope ($10^{-4}$kW$^{-1}$m$^2$) | $r^2$ | Comp. | Slope ($10^{-4}$kW$^{-1}$m$^2$ | $r^2$ | Comp. | Slope ($10^{-4}$kW$^{-1}$m$^2$) | $r^2$ |
| $C_9H_{12}O_2$ | 0.0799 | 0.6509 | $C_{15}H_{16}O_2$ | −3.0567 | 0.7644 | $C_8H_4O_6$ | −0.2049 | 0.8356 |
| $C_9H_{12}O_3$ | 0.3881 | 0.8997 | $C_{15}H_{16}O_3$ | −0.1781 | 0.7302 | $C_{10}H_8O_8$ | −0.3220 | 0.8041 |
| $C_9H_{12}O_4$ | 0.8524 | 0.8734 | $C_{15}H_{16}O_4$ | −0.1945 | 0.5876 | $C_{15}H_6O_8$ | −0.2841 | 0.9011 |
| $C_9H_{12}O_5$ | 1.2722 | 0.7881 | $C_{15}H_{16}O_5$ | −0.2408 | 0.2915 | $C_{16}H_6O_8$ | −0.2010 | 0.8574 |
| $C_9H_{12}O_6$ | 0.6680 | 0.7984 | $C_{15}H_{16}O_6$ | −0.0972 | 0.0141 | $C_{20}H_{12}O_5$ | −0.1691 | 0.7619 |
| $C_9H_{12}O_7$ | 0.1458 | 0.6259 | $C_{15}H_{16}O_7$ | −0.0528 | 0.0015 | $C_{20}H_{10}O_7$ | −0.1869 | 0.8595 |
| $C_9H_{12}O_8$ | −0.0333 | 0.2119 | $C_{15}H_{16}O_8$ | −0.1424 | 0.0137 | $C_{22}H_{12}O_9$ | −0.1910 | 0.9107 |

## 4. Discussion

From our experiments we are able to support or to reject the described hypotheses.

### 4.1. Differences in the Change of Dissolved Organic Matter (DOM) Composition (Suggested Presence/Absence of Bacteria) by Using Different Filter Sizes

To ensure that the analytical results of the experiment have been driven by photochemical processes, significant biological activities must be excluded as they were detected e.g., [16] in natural waters. Therefore, the experiment was performed on both 2 μm (presence of bacteria) and 0.2 μm (absence of bacteria) filtered water.

Assuming a bacterial growth efficiency of 10% within the experiment with 2 μm filtered water, only 1% of the depleted DOC can be explained (Figure 2e). The apparent low bacterial activity can be explained by the intense global irradiation, the higher temperature in the flasks in August 2018 than in the pre-dam water. Under these conditions bacteria had no chance to escape from the irradiation pressure. The upscaled measured value of the cumulated global radiation was comparable to the monthly sum for August 2018 of the DWD (Section 3.1) [75]. Similar values of global radiation was measured by [55] for arctic regions in the years 2011 and 2012. The bacteria were potentially damaged under these conditions showing negligible carbon processing activity. Under present natural conditions, for example in irradiated epilimnetic waters, the bacterial production should be considered. The experiment with sterile water, 0.2 μm filtered showed that there was no microbial activity. Consequently, the temporal behavior should not differ in both experimental series. The distribution of molecular formula derived Spearman's rank correlation coefficients $r_s$ was symmetrical to the diagonal line (y = x) suggesting that the correlations showed no evident difference in the behavior for both experiments (Figure S15). In summary, we have to reject our hypothesis i. In detail, there may be significant differences in some parameters at certain times (Table 1). The 2 μm experiment can be regarded as additional replicates.

The calculated BIX indicated also that this was a pure photochemical and not a biological process. A BIX >1 describes the proportion of autochthonously produced DOM of biological origin and values <0.6 allochthonous produced material (Figure 2d), according to [69]. From the viewpoint of BIX values we conclude for both experimental series that the produced material was not of biological origin. The initial FI of both series showed that the filtered water samples are not from microbial source, according to [39].

Due to the exclusion of biological processes in the experiment, the decrease of the DOC due to intensive irradiation was associated exclusively with photochemical processes. As shown in Figure 2a and Figure S7a, the DOC decreased during the experiment in both variants of the filter levels. The little difference after 4 h can be explained by the low bacterial production. Finally this means that the DOC decrease by 2.2 mg/L can be explained by a direct photo oxidation to $CO_2$ (8.1 mg/L), as a net process proportional to photon uptake, as described in [15].

### 4.2. Types of Photo-Reaction Behavior

With the experimental design we obtained detailed information about the behaviour of several parameters and especially of the molecular DOM components. Because of the high temporal resolution two important sudden drops during the photochemical process for both experiments could be detected. The first important observed point was at 83 kW/m$^2$ cumRAD, where some parameters like the HIX or several DOM components intensities decreased or increased. This means that overnight some dark reactions happened, which could not be explained by the experiment.

Another important point was at 168 kW/m$^2$ cumRAD. The sudden drop can be described by the necessary minimum activation energy for several reactions. The molecular formula $C_{15}H_{16}O_8$ is one example of an intermediate product, which increased overnight and decreased again at 168 kW/m$^2$ cumRAD (Figure 7b). This behavior was also identified for the SUVA$_{254}$, which was highly correlated

to $C_{10}H_8O_8$ (Figure S14). It can be concluded that overnight (dark period) and at 168 kW/m$^2$ cumRAD processes sharply change the molecular composition.

Some DOM peak intensities decreased or increased over the total experiment. Component $C_9H_{12}O_5$ was one of just 7 different molecular formulas, whose intensities increased over the experiment (Figure 7a). Other components intensities like that of $C_{15}H_6O_8$ decreased up to the end (Figure 7c). For this molecular formula only one structure was found in ChemSpider (5-hydroxy-1,4,8-trioxo-4,8-dihydro-1H-fluorene-3,9-dicarboxylic acid) and should suggest a chinoid system, which are often found in fluorescent molecules (Figure 7d).

We could verify hypothesis ii. There exist different types of photo reaction behavior. The reaction pathways may be much more complicated as derived from the 13 measured time points. Components like $C_9H_{12}O_5$ (Figure 7a) showed a monotonous intensity increase, components like $C_{15}H_{16}O_2$ (Figure 7b) monotonous intensity decrease. Components like $C_{15}H_{16}O_8$ (Figure 7b) showed first increasing and then decreasing intensity. From these examples, it becomes evident that, depending on the observed reaction time, components might be incorrectly assigned to products or degraded components, if only the start and endpoint intensities are measured (with sun simulator: [6,76,77], with natural sun: [78]. If the observation in our experiment would have been finished at t = 04 (83.3 kW/m$^2$), the component $C_{15}H_{16}O_5$ (Figure 7b) might have been identified as photo product. If the observation had been occurred only at start and end (436.2 kW/m$^2$), this component might have been identified as non-reactive or even as photo-degraded.

Our experiment provided elevated accuracy in the assignment of reaction types to DOM components.

*4.3. Average Photo-Chemical DOM Quality Change*

Photoproducts were related to relatively aliphatic and oxygen-poor components and to a smaller number of oxygen-rich and partly less aliphatic components as found by [76,79]. The degraded components included more unsaturated components, including the groups of lignin-likes and polyphenol-likes (Figures S1 and S2). In agreement with the loss of fluorescence the degree of aromaticity was lost on the first view (Table S7). The double bond equivalence (DBE) decreased as expected [9,80,81]. Nitrogen remains constant, carbon and oxygen decreased and sulphur increased, mostly from t = 10 (168 kW/m$^2$ cumRAD, second important drop). Probably, no sulphur was incorporated, but of what is left (carbon and oxygen), sulphur content is slightly higher. DOC was likely to be mineralized via decarboxylation. Oxygen was probably added to double bonds e.g., via oxidation, as an intermediate, but left the molecules also via decarboxylation. Both oxidation of double bonds and subsequent decarboxylation reactions can be driven by reactive species like OH radicals or singlet oxygen [54,82]. The average O/C kept nearly constant but H/C increased at the end of the experiment (from 168 kW/m$^2$ cumRAD) (Figures S1 and S2, Table S7). Accordingly, the molecules were reduced overall, which means the nominal oxidation state of carbon (NOSC) becomes more negative. The average mass was decreased until the end of the experiment (Table S7) as found also by [79].

The CHNO components (Figures S3 and S4), which were produced (red dots) are related to lignin-like components. N-containing photo products were also reported by [79,83]. The degraded components (black dots) included more condensed aromatics components. All CHNO components had a mass < 500 Da and were related to small molecules, with oxygen-poor and relatively aliphatic character (Figure S4). The photo transformation of S-containing components was investigated by [80] and [84]. In contrast to these two studies, the abundance of CHOS was much smaller in our sample (Figures S5 and S6).

The hypothesis iii was confirmed as follows. The photoproducts were more aliphatic (loss of DBE) and had less molecular weight (braking bonds, loss of CO$_2$ by decarboxylation) compared to photo degraded components. One suggestion of the hypothesis could be rejected: The photo products were not more oxygen-rich but less oxygen-rich as shown by the decreased average (intensity weighted) content of oxygen (Table S7). Evidently more oxygen was lost by decarboxylation compared to the

addition via oxidation of double bonds, which were clearly lost on average. This explains also the decreased NOSC (from −0.27 at the beginning of the experiment to −0.4 at the end of the experiment) by withdrawing highly oxidised carbon (+4) via decarboxylation.

*4.4. Identification of DOM Quality Changes with EEMs, SUVA and the Calculated PARAFAC Components*

　　The HIX, which represents a little part of the chromophoric dissolved organic matter (CDOM) [68], decreased overnight. At the beginning of the experiment initial HIX values of about 14 were observed. The HIX decreased to more than 80% in both experiment series (Table 1). That indicates removal of fluorescent aromatic material. This is confirmed in Figure 6: Components with H/C < 1 and O/C > 0.5 were photo degraded amongst others. These oxygen-rich and relatively unsaturated components (polyphenol-likes) were identified to be fluorescent [8,51] and they correlated positively to the HIX [52,72]. The component $C_{15}H_6O_8$ (Figure 7d) potentially matches such highly fluorescent and photo degradable substances. The photo products were more aliphatic in comparison to polyphenol-likes. Components with van Krevelen coordinates like the photo products in our experiment were found to be rather negatively correlated to the fluorescence intensity and the HIX [8,51,52,72]. This corroborates the strong decrease of fluorescence intensity and HIX as shown in Figures 2c and 3.

　　During the first 40 h, most of the high molecular weight and aromatic material was reduced. A selected portion of the fluorescence (HIX) decreased overnight, but an increase about 5% (peak C excitation (ex): 280 to 320 nm; emission (em): 380 to 460 nm) in fluorescence intensity could be identified by the formation of the difference spectra. Most of the fDOM was reduced (decrease of the fluorescence intensity of about 40–50%) on the first day. In total, there was a decrease of 50% for peak B (ex: 320 to 400 nm; em: 380 to 500 nm) and of 75% for peak A (ex: 240 to 280 nm; em: 380 to 500 nm). These two peaks were also relevant in [81] or [85], where they also decreased under UV irradiation. This means that also the fDOM was reduced, more than the chromophoric DOM. The component $C_{15}H_{16}O_2$ (light green curve in Figure 7b) correlates highly with the HIX (Figure S13b).

　　The high initial SUVA$_{254}$ value of about 4 L mg $C^{-1}$ $m^{-1}$ reflects that the DOM in the Rote Mulde included high molecular weight and aromatic compounds, which were strongly unsaturated and mainly hydrophobic [62]. At the sudden drop at 168 kW/m$^2$ cumRAD, the SUVA values decreased below 3, indicating that the material got a hydrophilic character [62] and high aromatic components were continuously converted or transformed into smaller molecules (low molecular weight) at the end. The correlation of the first PARAFAC component with SUVA$_{254}$ was also shown by [86].

　　The first PARAFAC component showed the drop overnight but rather correlated with the SUVA (Figure S14a) and the second component was correlated to the HIX (Figure S14b), which showed that the second component was more photochemically degradable in comparison to the first component. The first component is described to be of terrestrial origin, absent in wastewater and with high concentrations in e.g., forest streams and the second component is humic-like and as present in all environments [65]. The total fit of the model was only about 50%. One reason could be the reduced fluorescence intensity within the pure photochemical experiment.

　　Other studies like [9,85,87] found similar PARAFAC components and the decrease of characteristic regions during irradiation with sunlight. We can confirm hypothesis iv and showed, that fluorescent components, in particular poly-phenol likes [8,51,52,72], were photo degraded.

## 5. Conclusions

　　In the photochemical experiment, the degradation of DOC to $CO_2$ was about 1/3 at cumulated global radiation of 436 kW/m$^2$. The holistic approach of combining information of the sum parameters (SUVA$_{254}$ and HIX) and several DOM components with the cumulated global radiation gives a deeper insight into the system dynamics and made it possible to observe two interesting points (overnight: 84 KW/m$^2$ and at 168 kW/m$^2$) where molecular composition drastically.

The built pre-dam influences the behavior of the humic-rich water. DOM, which comes from forested landscape, showed high potential for transformation and degradation when suddenly exposed to sunlight and had higher DOC concentrations [2–4]. This could be confirmed by this experiment.

Because of the semi-natural experiment we could not investigate the bacterial production. Future work should therefore carry out this experiment within the pre dam to check how the bacterial production behaves and how deep the global radiation gets into the surface of the water. Also the different temperature can have an effect on the photo products and the bacterial production. It must be investigated, whether similar photo products are found under natural conditions or whether other components arise when bacterial production increases.

**Supplementary Materials:** The following are available online at http://www.mdpi.com/2073-4441/12/2/331/s1: SI.docx: additional van Krevelen diagrams broken down to component classes, additional diagrams with data for the 2 μm filtered water, EEMF spectra showing PARAFAC components, results of split half validation, all fluorescence difference spectra, scores of PARAFAC components, comparison of $r_s$ values 0.2 μm versus 2 μm filtration, time table for the experiment procedure, list of fluorescence indices values, SI_data_base.xlsx: elemental formula components data with calculated slopes and $r_s$ values, intensity weighted average parameters.

**Author Contributions:** Conceptualization, C.W., W.v.T. and P.H.; methodology, C.W., P.H., O.J.L., N.K. and W.v.T.; software, C.W. and O.J.L.; validation, C.W. and W.v.T.; formal analysis, C.W., P.H., O.J.L. and N.K.; investigation, C.W. and P.H.; data curation, C.W. and P.H.; writing—original draft preparation, C.W.; writing—review and editing, C.W., P.H., N.K., O.J.L. and W.v.T.; visualization, C.W., P.H. and N.K.; supervision, W.v.T., P.H. and N.K. All authors have read and agreed to the published version of the manuscript.

**Funding:** The PhD scholarship was founded by the German Federal Environmental Foundation (Deutsche Bundesstiftung Umwelt—DBU).

**Acknowledgments:** We thank the laboratory team (GEWANA), especially Ines Locker and Kerstin Lerche for her help during the field sampling and the extractions for the FT-ICR MS analysis and also Ina Siebert for her help in the lab with the carbon concentration analysis and for the sampling during the whole experiment. We also want to thank Kai Franze for software development and Jan M. Kaesler for the FT-ICR MS analysis at the Centre for Chemical Microscopy (ProVIS) at the Helmholtz Centre for Environmental Research which is supported by European Regional Development Funds (EFRE-Europe funds Saxony) and the Helmholtz Association. Furthermore, we would like to thank Prof. Dr. rer. nat. habil. Jürgen W. Einax from the Friedrich-Schiller-University for his helpful comments. We thank two anonymous reviewers for their helpful comments.

**Conflicts of Interest:** The authors declare no conflict of interest. The funders had no role in the design of the study; in the collection, analyses, or interpretation of data; in the writing of the manuscript; or in the decision to publish the results.

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
