# Peer review of "Photochemically Induced Changes of Dissolved Organic Matter in a Humic-Rich and Forested Stream"

_water, doi:10.3390/w12020331_

Round 1
Reviewer 1 Report
In this manuscript, the authors investigate the effects of photooxidation of DOM sampled from a forested stream using EEM/PARAFAC and ICR-MS. The study was conducted in a very rigorous manner and documented well to allow the reader a good understanding of how the experiment was carried out. The study shows the importance of having sufficient sampling points to capture transformations that would not be evident with less frequent sampling.
The manuscript is clearly written and the results and discussion are sound. The only significant criticism that I have is that the ICR-MS results are based on an intensity-form of the mass spectra. While technically a qualitative method that perhaps in its purest form should be analyzed by a binary presence/absence approach, there are times when, with caution, intensity based approaches are fruitful in furthering the understanding of the chemical details of the ecosystem process under investigation.
I think the authors need to consider how the changing composition due to oxidation may affect the intensities of the peaks due to competition ionization effects. Especially in the case of oxidation reactions, I suspect that some of the changes in intensities are due to competitive effects, rather than "treatment, i.e. photo-oxidation" effects. The authors should add a short paragraph of this potential issue in the interpretation of the results.
Just a minor suggestion is that the abstract does not mention ICR results despite it playing a major role in the data.
Author Response
Response letter attached.

Reviewer 2 Report
Review of a manuscript submitted to the Water, entitled “Photochemically induced changes of dissolved organic matter in a humic-rich and forested stream”, by Christin Wilske, Peter Herzsprung , Oliver J. Lechtenfeld , Norbert Kamjunke , Wolf Von Tümpling.
GENERAL COMMENTS
Photochemical processing is an important way of the transformation of terrestrial dissolved organic matter (DOM) but was rarely investigated by ultra-high resolution mass spectrometry. Land uses including included different forest types affect the amount, quality and sources of DOM in streams and rivers. This study shows the molecular change of terrestrial DOM before the preparation of drinking water from reservoirs. The substantive content and formal content of the article does not raise any objections. The article does meets the requirements the formal recommendations of "Instructions for Authors", the is follow the PRISMA. The organization of the article is a satisfactory. The content justify the length of the article. I think that the article is interesting and fully deserves to be published. The subject of the paper fall within the scope of the journal. The title of this paper clearly reflect its content. It is a new and original contribution in research on photochemically induced changes of dissolved organic matter in a humic-rich and forested stream. The keywords informative and appropriate. The abstract sufficiently informative especially when read in isolation, with a slight doubt as to:
Paragraph line 78-101 I suggest moving to the "Materials and Methods" chapter (2.8.2 Statistical analysis of bulk parameters/2.8.4 Differentiation between photo products and photo degraded components).
The statement of objectives of the paper is adequate and appropriate in view of the subject matter. The methods exposed correctly and sufficiently informative to allow replications of the research. The statistical methods used correctly and adequate. The illustrations and tables all necessary, complete, clearly presented, and are the captions adequate and informative. The interpretations and conclusions sound, justified by the data and consistent with the objectives. Analyzing the results and discussing the support or rejection of hypotheses is properly documented and convincing.
Author Response
Response letter attached.
